# Continual Task Learning through Adaptive Policy Self-Composition

## Abstract

Training a generalizable agent to continually learn a sequence of tasks from offline trajectories is a natural requirement for long-lived agents, yet remains a significant challenge for current offline reinforcement learning (RL) algorithms. Specifically, an agent must be able to rapidly adapt to new tasks using newly collected trajectories (plasticity), while retaining knowledge from previously learned tasks (stability). However, systematic analyses of this setting are scarce, and it remains unclear whether conventional continual learning (CL) methods are effective in continual offline RL (CORL) scenarios. In this study, we develop the Offline Continual World benchmark and demonstrate that traditional CL methods struggle with catastrophic forgetting, primarily due to the unique distribution shifts inherent to CORL scenarios. To address this challenge, we introduce CompoFormer, a structure-based continual transformer model that adaptively composes previous policies via a meta-policy network. Upon encountering a new task, CompoFormer leverages semantic correlations to selectively integrate relevant prior policies alongside newly trained parameters, thereby enhancing knowledge sharing and accelerating the learning process. Our experiments reveal that CompoFormer outperforms conventional CL methods, particularly in longer task sequences, showcasing a promising balance between plasticity and stability.

## 1 Introduction

Similar to human cognition, a general-purpose intelligent agent is expected to continually acquire new tasks. These sequential tasks can be learned either through online exploration (Malagon et al., 2024; Yang et al., 2023; Wolczyk et al., 2021) or offline from pre-collected datasets (Huang et al., 2024; Gai et al., 2023; Hu et al., 2024a), with the latter being equally critical but having received comparatively less attention. Furthermore, online learning is not always feasible due to the need for direct interaction with the environment, which can be prohibitively expensive in real-world settings. Thus, the study of continual offline reinforcement learning (CORL) is both crucial and valuable for advancing general-purpose intelligence, which integrates offline RL and continual learning (CL).

Although offline RL has demonstrated strong performance in learning single tasks (Chen et al., 2021; Hu et al., 2024b;c), it remains prone to catastrophic forgetting and cross-task interference when applied to sequential task learning (Bengio et al., 2020; Fang et al., 2019). Moreover, there is a lack of systematic analysis in this setting, and it remains unclear whether conventional CL methods are effective in CORL settings. CORL faces unique challenges, including distribution shifts between the behavior and learned policies, across offline data from different tasks, and between the learned policy and saved replay buffers (Gai et al., 2023). As a result, managing the stability-plasticity tradeoff in CL and addressing the challenges posed by distribution shifts in offline RL remains a critical issue (Yue et al., 2024). To tackle this, we leverage sequence modeling using the Transformer architecture (Hu et al., 2024d; Vaswani, 2017), which reformulates temporal difference (TD) learning into a behavior cloning framework, enabling a supervised learning approach that aligns better with traditional CL methods. While Transformer-based methods help reduce distribution shift between the behavior and learned policies and facilitate knowledge transfer between similar tasks due to the model's strong memory capabilities (Chen et al., 2021), they may exacerbate shifts between time-evolving datasets and between the learned policy and replay buffer (Huang et al., 2024). Consequently, these methods still face significant stability-plasticity tradeoff problems, even when combined with CL approaches for learning sequences of new tasks (see Section 5.2 for details).

Figure 1: Adaptive policy self-composition architecture. When a new task arises (represented by a star), its textual description is processed by the frozen S-BERT model to compute attention scores with previous task descriptions. After several update iterations of the attention module, if the composed policy is sufficient for the current task, it is used directly; otherwise, new parameters are incorporated alongside the composed policy to construct the new policy $\pi^{(k)}$.

To overcome the limitations of previous approaches and better address the plasticity-stability trade-off, we explore how to train a meta-policy network in CORL. Structure-based methods typically introduce new sub-networks for each task while preserving parameters from previous tasks, thereby mitigating forgetting and interference (Mallya & Lazebnik, 2018). These methods transfer knowledge between modules by transferring relevant knowledge through task-shared parameters. However, as the number of tasks increases, distinguishing valuable information from shared knowledge becomes increasingly challenging (Malagon et al., 2024). In such cases, knowledge sharing may offer limited benefits and could even impede learning on the current task. Therefore, in this paper, we propose leveraging semantic correlations to selectively compose relevant prior learned policy modules, thereby enhancing knowledge sharing and accelerating the learning process.

As illustrated in Figure 1, when a new task is introduced, our model, CompoFormer, first utilizes its textual description to compute attention scores with descriptions of previous tasks using the frozen Sentence-BERT (S-BERT) (Reimers, 2019) module and a trainable attention module. After several update iterations based on the final output action loss, if the composed policy output is sufficient for the current task, it is applied directly. Otherwise, new parameters are integrated alongside the composed policy to construct a new policy for the new task. This process naturally forms a cascading structure of policies, growing in depth as new tasks are introduced, with each policy able to access and compose outputs from prior policies to address the current task (Malagon et al., 2024). This approach could effectively manage the plasticity-stability trade-off. Specifically, relevant tasks share a greater amount of knowledge, facilitating a faster learning process, while less knowledge is shared between unrelated tasks to minimize cross-task interference. This mechanism enhances plasticity and accelerates the learning process. Simultaneously, the parameters of previously learned tasks remain fixed, preserving stability and reducing the risk of forgetting.

In our experiments, we extend the Continual World benchmark (Wolczyk et al., 2021) to the Offline Continual World benchmark and conduct a comprehensive evaluation of existing CL methods. Our approach consistently outperforms these baselines across different benchmarks, particularly in longer task sequences, achieving a marked reduction in forgetting (Table 1). This demonstrates its ability to effectively leverage prior knowledge and accelerate the learning process by adaptively selecting relevant policies. Additionally, we highlight the critical contributions of each component in our framework (Figure 4) and show the generalizability of our method across varying task order sequences (Table 2). Compared to state-of-the-art approaches, CompoFormer achieves the best trade-off between plasticity and stability (Figure 5).

## 2 RELATED WORK

**Offline RL.** Offline RL focuses on learning policies solely from static, offline datasets $\mathcal{D}$, without requiring any further interaction with the environment (Levine et al., 2020). This paradigm signif-

icantly enhances the sample efficiency of RL, particularly in scenarios where interacting with the environment is costly or involves considerable risk, such as safety-critical applications. However, a key challenge in offline RL arises from the *distribution shift* between the learned policy and the behavior policy that generated the dataset, which can lead to substantial performance degradation (Fujimoto et al., 2019). To address this issue, various offline RL algorithms employ constrained or regularized dynamic programming techniques to minimize deviations from the behavior policy and mitigate the impact of distribution shift (Fujimoto & Gu, 2021; Kumar et al., 2020; Kostrikov et al., 2021). Another promising approach for offline RL is conditional sequence modeling (Hu et al., 2024d; Chen et al., 2021), which predicts future actions based on sequences of past experiences, represented by state-action-reward triplets. This approach naturally aligns with supervised learning, as it restricts the learned policy to remain within the distribution of the behavior policy, while being conditioned on specific metrics for future trajectories (Hu et al., 2023a; 2024b; Yamagata et al., 2023; Hu et al., 2023b; 2024e; Meng et al., 2023). Given that most continual learning methods follow a supervised learning framework, we adopt the Decision Transformer (Chen et al., 2021) as the base model and implement various continual learning techniques in conjunction with it.

**Continual RL.** Continual learning is a critical and challenging problem in machine learning, aiming to enable models to learn from a continuous stream of tasks without forgetting previously acquired knowledge. Generally, continual learning methods can be categorized into three main approaches (Masana et al., 2022): (i) Regularization-based approaches (Kirkpatrick et al., 2017; Aljundi et al., 2018), which introduce regularization terms to prevent model parameters from drifting too far from those learned from prior tasks; (ii) Structure-based methods (Mallya & Lazebnik, 2018; Huang et al., 2024), which allocate fixed subsets of parameters to specific tasks; (iii) Rehearsal-based methods (Chaudhry et al., 2018b; Wolczyk et al., 2021), which retrain the model by merging a small amount of data from previously learned tasks with data from the current task. Most of these methods have been extensively investigated within the context of online RL (Yang et al., 2023; Malagon et al., 2024), whereas there is a paucity of research focused on adapting them to offline RL settings (Huang et al., 2024; Gai et al., 2023). Furthermore, these efforts often differ in their experimental settings, evaluation metrics, and primarily focus on rehearsal-based methods, often leveraging diffusion models (Hu et al., 2024a). Nonetheless, they continue to grapple with the significant stability-plasticity dilemma (Khetarpal et al., 2022). In this paper, we introduce the Offline Continual World benchmark to perform a comprehensive empirical evaluation of existing continual learning methods. Our approach demonstrates a promising balance between plasticity and stability.

## 3 PRELIMINARIES

We begin by introducing the notation used in the paper and formalizing the problem at hand.

The goal of RL is to learn a policy $\pi_\theta(\mathbf{a}|\mathbf{s})$ maximizing the expected cumulative discounted rewards $\mathbb{E}[\sum_{t=0}^{\infty} \gamma^t \mathcal{R}(\mathbf{s}_t, \mathbf{a}_t)]$ in a Markov decision process (MDP), which is a six-tuple $(\mathcal{S}, \mathcal{A}, \mathcal{P}, \mathcal{R}, \gamma, d_0)$, with state space $\mathcal{S}$, action space $\mathcal{A}$, environment dynamics $\mathcal{P}(\mathbf{s}'|\mathbf{s}, \mathbf{a}) : \mathcal{S} \times \mathcal{S} \times \mathcal{A} \rightarrow [0, 1]$, reward function $\mathcal{R} : \mathcal{S} \times \mathcal{A} \rightarrow \mathbb{R}$, discount factor $\gamma \in [0, 1)$, and initial state distribution $d_0$ (Sutton & Barto, 2018). In the offline setting (Levine et al., 2020), instead of the online environment, a static dataset $\mathcal{D} = \{(\mathbf{s}, \mathbf{a}, \mathbf{s}', r)\}$, collected by a behavior policy $\pi_\beta$, is provided. Offline RL algorithms learn a policy entirely from this static offline dataset $\mathcal{D}$, without online interactions with the environment.

In this work, we follow the task-incremental setting commonly used in prior research (Khetarpal et al., 2022; Wolczyk et al., 2021), where non-stationary environments are modeled as MDPs with components that may vary over time. A task $k$ is defined as a stationary MDP $\mathcal{M}^{(k)} = \langle \mathcal{S}^{(k)}, \mathcal{A}^{(k)}, \mathcal{P}^{(k)}, \mathcal{R}^{(k)}, \gamma^{(k)}, d_0^{(k)} \rangle$, where $k$ is a discrete index that changes over time, forming a sequence of tasks. For each task, a corresponding static dataset $\mathcal{D}^{(k)} = \{(\mathbf{s}^{(k)}, \mathbf{a}^{(k)}, \mathbf{s}'^{(k)}, r^{(k)})\}$ is collected by a behavior policy $\pi_\beta^{(k)}$. We assume that the agent is provided with a limited budget of training steps to optimize the task-specific policy $\pi^{(k)}$ using the dataset $\mathcal{D}^{(k)}$. Once this budget is exhausted, a new task $\mathcal{M}^{(k+1)}$ is introduced, and the agent is restricted to interacting solely with this new task. The objective is to accelerate and improve the optimization of the policy $\pi^{(k)}$ for the current task $\mathcal{M}^{(k)}$ by leveraging knowledge from the previously learned policies $\{\pi^{(i)}\}_{i=1,...,k-1}$.

We adopt three common assumptions regarding variations between tasks, following prior work (Wolczyk et al., 2021; Khetarpal et al., 2022; Malagon et al., 2024). First, the action space $\mathcal{A}$ remains

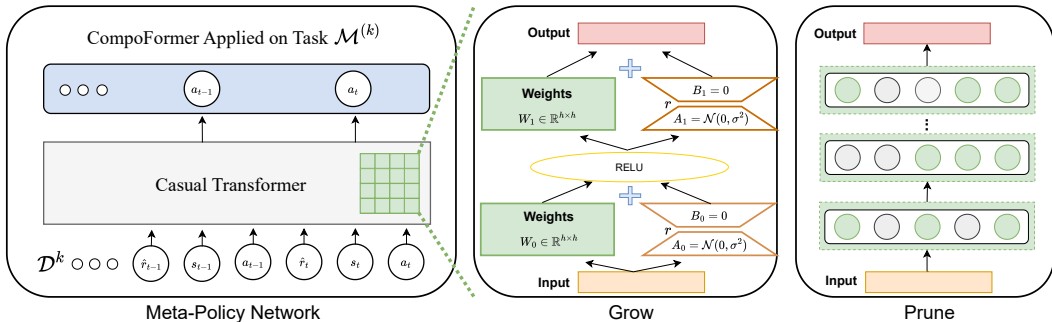

Figure 2: The architecture of the meta-policy network. The core module is built upon the Transformer architecture, which receives the trajectory as input and outputs the corresponding action. Upon encountering task $\mathcal{M}^{(k)}$, our method presents two variants: the "Grow" variant, which adds new parameters to the Transformer in a LoRA format, and the "Prune" variant, which utilizes a masking technique to deactivate certain parameters within the Transformer (where green indicates activated parameters and grey denotes inactivated ones).

constant across all tasks. This assumption is flexible, as tasks with distinct action sets can be treated as sharing a common action space $\mathcal{A}$ by assigning a zero probability to task-irrelevant actions. Second, task transition boundaries and identifiers are assumed to be known to the agent, as is standard in the literature (Wolczyk et al., 2021; 2022; Khetarpal et al., 2022). Finally, variations between tasks primarily arise from differences in environment dynamics ($\mathcal{P}$) and reward functions ($\mathcal{R}$), implying that tasks should have similar state spaces, i.e., $\mathcal{S}^{(i)} \approx \mathcal{S}^{(j)}$.

## 4 METHOD

In this work, given a sequence of previously learned policies $\{\pi^{(i)}\}_{i=1,\dots,k-1}$, where each policy $\pi^{(i)}$ corresponds to task $\mathcal{M}^{(i)}$, we identify two scenarios for the current task $\mathcal{M}^{(k)}$: (i) $\mathcal{M}^{(k)}$ can be solved by a previously learned policy without requiring additional parameters, or (ii) $\mathcal{M}^{(k)}$ requires learning a new policy that leverages knowledge from previous tasks to accelerate learning without interfering with earlier policies. Our methods address this by allocating a sub-network for each task and freezing its weights after training, effectively preventing forgetting. There are two main approaches to sub-network representation: (i) adding new parameters to construct the sub-network (Czarnecki et al., 2018; Malagon et al., 2024; Huang et al., 2024), or (ii) applying binary masks to neurons' outputs (Serra et al., 2018; Ke et al., 2021; Mallya & Lazebnik, 2018). We provide solutions for both approaches. The detailed pipeline is summarized in Algorithm 1.

### 4.1 META-POLICY NETWORK

The underlying network architecture is based on the Decision Transformer (DT) (Chen et al., 2021), which formulates the RL problem as sequence modeling, leveraging the scalability of the Transformer architecture and the benefits of parameter sharing to better exploit task similarities.

Specifically, during training on offline data for the current task $\mathcal{M}^{(k)}$, DT takes the recent $M$-step trajectory history $\tau_t^{(k)}$ as input, where $t$ represents the timestep. This trajectory consists of the state $\mathbf{s}_t^{(k)}$, the action $\mathbf{a}_t^{(k)}$, and the return-to-go $\hat{R}_t^{(k)} = \sum_{i=t}^{T} r_i^{(k)}$, where $T$ is the maximum number of interactions with the environment. The trajectory is formulated as:

$$\tau_t^{(k)} = (\hat{R}_{t-M+1}^{(k)}, \mathbf{s}_{t-M+1}^{(k)}, \mathbf{a}_{t-M+1}^{(k)}, \dots, \hat{R}_t^{(k)}, \mathbf{s}_t^{(k)}, \mathbf{a}_t^{(k)}). \tag{1}$$

The prediction head linked to a state token $\mathbf{s}_t^{(k)}$ is designed to predict the corresponding action $\mathbf{a}_t^{(k)}$. For continuous action spaces, the training objective aims to minimize the mean-squared loss:

$$\mathcal{L}_{DT} = \mathbb{E}_{\tau_t^{(k)} \sim \mathcal{D}^{(k)}} \left[ \frac{1}{M} \sum_{m=t-M+1}^{t} (\mathbf{a}_m^{(k)} - \pi(\tau_t^{(k)})_m)^2 \right]. \tag{2}$$

where $\pi(\tau_t^{(k)})_m$ is the $m$-th action output of the Transformer policy $\pi$ in an auto-regressive manner.

Based on the construction of the new policy sub-network, our method can be divided into two variants: CompoFormer-Grow and CompoFormer-Prune (Shown in Figure 2).

**CompoFormer-Grow.** When a new sub-network needs to be added, its architecture is expanded by introducing additional parameters. Rather than adding fully linear layers, we employ Low-Rank Adaptation (LoRA) (Hu et al., 2021), which integrates rank-decomposition weight matrices (referred to as update matrices) into the existing network. Only these newly introduced parameters are trainable, enabling efficient fine-tuning while minimizing forgetting. Specifically, the form of the LoRA-based multilayer perceptron (LoRA-MLP) is expressed as:

$$\text{LoRA-MLP}(\boldsymbol{X}) = (\boldsymbol{W}_1 + \boldsymbol{B}_1\boldsymbol{A}_1)(\text{RELU}((\boldsymbol{W}_0 + \boldsymbol{B}_0\boldsymbol{A}_0)\boldsymbol{X} + \boldsymbol{b}_0)) + \boldsymbol{b}_1, \tag{3}$$

where $\boldsymbol{W}_0 \in \mathbb{R}^{h \times h}$ and $\boldsymbol{W}_1 \in \mathbb{R}^{h \times h}$ are the weights of two linear layers, $\boldsymbol{b}$ is the bias, and $\boldsymbol{A}_0 \in \mathbb{R}^{r \times h}, \boldsymbol{B}_0 \in \mathbb{R}^{h \times r}, \boldsymbol{A}_1 \in \mathbb{R}^{r \times h}$ and $\boldsymbol{B}_1 \in \mathbb{R}^{h \times r}$ are the update matrices. Here, $r$ signifies the rank of LoRA, $h$ indicates the hidden dimension, and it holds that $r << h$. For the first task $\mathcal{M}^{(1)}$, we train all parameters of the DT model. For subsequent tasks $\mathcal{M}^{(k)}$ where $k > 1$, only the newly introduced parameters, $L \times [\boldsymbol{A}_0^{(k)}, \boldsymbol{B}_0^{(k)}, \boldsymbol{A}_1^{(k)}, \boldsymbol{B}_1^{(k)}]$, are updated, where $L$ denotes the number of hidden layers in the DT.

**CompoFormer-Prune.** Given a meta-policy network with $L$ layers, where the final layer $L$ serves as the action prediction head, let $l \in \{1, \dots, L-1\}$ index the hidden layers. In this context, $\boldsymbol{y}_l$ represents the output vector of layer $l$, and $\theta_l$ denotes the weights of layer $l$. The output of the sub-network at layer $(l+1)$ is:

$$\boldsymbol{y}_{l+1} = f(\boldsymbol{y}_l; \phi_{l+1}^{(k)} \otimes \theta_{l+1}), \tag{4}$$

where $\phi_{l+1}^{(k)}$ is a binary mask generated for task $\mathcal{M}^{(k)}$ and applied to layer $l+1$, and $f$ represents the neural operation (e.g., a fully connected layer). The element-wise product operator $\otimes$ activates a subset of neurons in layer $(l+1)$ according to $\phi_{l+1}^{(k)}$. These activated neurons across all layers form a task-specific sub-network, which is updated with the offline dataset.

To efficiently utilize the network's capacity, each task-specific policy should form a sparse sub-network, activating only a small subset of neurons. Inspired by Mallya & Lazebnik (2018), during training on task $\mathcal{M}^{(k)}$, we first employ the full network with parameters $\theta_l$ across all layers $l \in \{1, 2, \dots, L\}$ to compute actions. The parameters are then updated without interfering with previously learned tasks by setting the gradients of $\theta_l \otimes \phi_l^{(k-1)}$ to zero in all layers. After several iterations of training, a fraction of the unused parameters from previous tasks, $\theta_l \otimes (1 - \phi_l^{(k-1)})$, are pruned—i.e., set to zero based on their absolute magnitude. The remaining selected weights, $\theta_l \otimes \phi_l'$, are then retrained to prevent performance degradation following pruning. This creates the new sub-network for task $\mathcal{M}^{(k)}$, with the updated mask defined as $\phi_l^{(k)} = \phi_l^{(k-1)} \mid \phi_l'$ across all layers, allowing for the reuse of weights from previous tasks. This process is repeated until all required tasks are incorporated or no free parameters remain.

## 4.2 Self-Composing Policy Module

In this subsection, we describe the key building block of the proposed architecture: the self-composing policy module. The following lines explain the rationale behind these components.

To expedite knowledge transfer from previous tasks to new tasks, we utilize the task's textual description with the aid of Sentence-BERT (S-BERT) (Reimers, 2019). Specifically, given a new task with $\mathcal{M}^{(k)}$ an associated description, we process the text using a pretrained S-BERT model to produce a task embedding $\boldsymbol{e}_k \in \mathbb{R}^d$, where $d$ denotes the output dimension of the S-BERT model. Then we introduce an attention module designed to adaptively learn the relevance of previously learned policies to the new task based on semantic correlations. Specifically, the query vector is computed as $\boldsymbol{q} = \boldsymbol{e}_k\boldsymbol{W}^Q$, where $\boldsymbol{W}^Q \in \mathbb{R}^{d \times h}$. As illustrated in Figure 2, the keys are computed as $\boldsymbol{K} = \boldsymbol{E}\boldsymbol{W}^K$, where $\boldsymbol{E}$ is the row-wise concatenation of task embeddings from prior tasks, and $\boldsymbol{W}^K \in \mathbb{R}^{d \times h}$. For the values matrix, no linear transformation is applied, instead, $\boldsymbol{V}$ is the row-wise concatenation of output features from previous policies $\{\Phi^{(1:k-1)}\}$, denoted by $\boldsymbol{V} = ||_{i=1}^{k-1} \Phi^{(i)}$, where $||$ denotes concatenation, and $\Phi^{(i)}$ represents the output feature of the newly constructed sub-network for task $\mathcal{M}^{(i)}$. Once $\boldsymbol{q}, \boldsymbol{K}$ are computed, the output of this block is obtained by the scaled

dot-product attention (Vaswani, 2017), as formulated as:

$$F(\boldsymbol{e}_k, \{\Phi^{(1:k-1)}\}) = \text{Attention}(\boldsymbol{q}, \boldsymbol{K}, \boldsymbol{V}) = \text{softmax}(\frac{\boldsymbol{q}\boldsymbol{K}^T}{\sqrt{d}})\boldsymbol{V}, \quad (5)$$

where $F$ represents the function of the attention module, and the learnable parameters in this block are $\boldsymbol{W}^Q$ and $\boldsymbol{W}^K$.

When encountering a new task $\mathcal{M}^{(k)}$, we first assign a set of parameters $\{\boldsymbol{W}^Q, \boldsymbol{W}^K\}^{(k)}$ and update them to evaluate whether previous tasks are capable of solving the current task, based on a predefined threshold. If the threshold is not met, new architecture parameters are assigned according to the method being used: in the case of CompoFormer-Grow, a new set of LoRA parameters is introduced, while for CompoFormer-Prune, idle parameters are pruned for the new task. To fully integrate prior knowledge, the output feature of the newly constructed sub-network, $\Phi^{(k)}$, is concatenated with the output of the attention module to generate the final action output, followed by an MLP layer:

$$\pi^{(k)} = \text{MLP}(\Phi^{(k)} \,||\, F(\boldsymbol{e}_k, \{\Phi^{(1:k-1)}\})), \quad (6)$$

where $||$ denotes concatenation. A detailed analysis of the computational cost of inference for our method is provided in Appendix D.

At the beginning of Section 4, we outlined two scenarios concerning the current task and previously learned policy modules. The following lines review these scenarios within the proposed architecture:

(i) If a previous policy is capable of solving the current task, the attention mechanism assigns high importance to it, avoiding the need for additional learnable parameters. This simplifies both the learning process and computational requirements.

(ii) When previous policies are insufficient to solve the current task, new parameters are introduced, and their output is concatenated with the output of the attention module to facilitate effective knowledge transfer and accelerate the learning process.

## 5 EXPERIMENT

### 5.1 EXPERIMENTAL SETUPS

**Benchmarks.** To evaluate CompoFormer, we introduce the Offline Continual World (OCW) benchmark, built on the Meta-World framework (Yu et al., 2020), to conduct a comprehensive empirical evaluation of existing continual learning methods. Specifically, we replicate the widely used Continual World (CW) framework (Wolczyk et al., 2021) in the continual RL domain by constructing 10 representative manipulation tasks with corresponding offline datasets. To increase the benchmark's difficulty, tasks are ranked according to a pre-computed transfer matrix, ensuring significant variation in forward transfer both across the entire sequence and locally (Yang et al., 2023). Additionally, we employ OCW20, which repeats the OCW10 task sequence twice, to evaluate the transferability of learned policies when the same tasks are revisited.

**Evaluation metrics.** Following a widely-used evaluation protocol in the continual learning literature (Rolnick et al., 2019; Chaudhry et al., 2018b; Wolczyk et al., 2021; 2022), we adopt three key metrics: (1) *Average Performance* (higher is better): The average performance at time $t$ is defined as $AP(t) = \frac{1}{K}\sum_{i=1}^{K} p_i(t)$ where $p_i(t) \in [0, 1]$ denotes the success rate of task $i$ at time $t$. This is a canonical metric used in the continual learning community. (2) *Forgetting* (lower is better): This metric quantifies the average performance degradation across all tasks at the end of training, denoted by $F = \frac{1}{K}\sum_{i=1}^{K} p_i(i \cdot \delta) - p_i(K \cdot \delta)$, where $\delta$ represents the allocated training steps for each task. (3) *Forward Transfer* (higher is better): This metric measures the impact that learning a task has on the performance of future tasks (Lopez-Paz & Ranzato, 2017), and is denoted by $FWT = \frac{1}{K-1}\sum_{i=1}^{K} p_i((i-1) \cdot \delta) - p_i(0)$.

**Comparing methods.** We compare CompoFormer against several baselines and state-of-the-art (SoTA) continual RL methods. As outlined by Masana et al. (2022), these methods can be categorized into three groups: regularization-based, structure-based, and rehearsal-based approaches. Concretely, the regularization-based methods include L2, Elastic Weight Consolidation (EWC) (Kirkpatrick et al., 2017), MemoryAware Synapses (MAS) (Aljundi et al., 2018), Learning without For-

---

**Algorithm 1** CompoFormer

---

1: **Initialize:** meta-policy network $\pi$, performance threshold $\eta$, $flag = True$.
2: **Input:** training budget $I_{tb}$, warmup budget $I_{wp}$, a sequence of tasks $\{\mathcal{M}^{(i)}\}_{i=1}^{K}$ with corresponding offline dataset $\{\mathcal{D}^{(i)}\}_{i=1}^{K}$.
3: **for** $k = 1$ to $K$ **do**
4:     Assign a new head for task $k$: head$^{(k)}$.
5:     Compute task embedding $e_k = f_{\text{S-BERT}}$(textual description of task $k$).
6:     **if** $k > 1$ **then**
7:         `// Determine if new parameters are needed.`
8:         Assign parameters $\{\boldsymbol{W}^Q, \boldsymbol{W}^K\}^{(k)}$.
9:         **for** $i = 1$ to $I_{wp}$ **do**
10:           Sample trajectory $\tau^{(k)}$ from $\mathcal{D}^{(k)}$.
11:           Compute loss with Equation 2 using output feature from Equation 5 and head$^{(k)}$.
12:           Update $\{\boldsymbol{W}^Q, \boldsymbol{W}^K\}^{(k)}$ and the parameters of head$^{(k)}$.
13:         **end for**
14:         Evaluate task $k$ with previous policies and learned attention module. Set $flag = False$ if performance $\geq \eta$, otherwise $flag = True$.
15:     **end if**
16:     **if** $flag$ is $True$ **then**
17:         `// Add new parameters`
18:         Add LoRA parameters for CompoFormer-Grow or prune idle parameters for CompoFormer-Prune to construct the new sub-network $\Phi^{(k)}$.
19:         **for** $i = 1$ to $I_{tb}$ **do**
20:           Sample trajectory $\tau^{(k)}$ from $\mathcal{D}^{(k)}$.
21:           Compute action output with Equation 6.
22:           Compute loss with Equation 2 and update corresponding parameters.
23:         **end for**
24:     **end if**
25: **end for**

---

getting (LwF) (Li & Hoiem, 2017), Riemanian Walk (RWalk) (Chaudhry et al., 2018a), and Variational Continual Learning (VCL) (Nguyen et al., 2017). Structure-based methods include PackNet (Mallya & Lazebnik, 2018) and LoRA (Huang et al., 2024). Rehearsal-based methods encompass Perfect Memory (PM) (Wolczyk et al., 2021) and Average Gradient Episodic Memory (A-GEM) (Chaudhry et al., 2018b). Additionally, we include the naive sequential training method (Finetuning) and the multi-task RL baseline MTL (Hu et al., 2024c), which are typically regarded as the soft upper bound for continual RL methods. A more detailed description and discussion of these methods, along with training details, are provided in Appendix B and C.

## 5.2 MAIN RESULTS

Table 1 and Figure 3 present the results of all methods evaluated under two settings, OCW10 and OCW20, using the metrics outlined in Section 5.1.

Both regularization-based and rehearsal-based methods struggle to perform well, in contrast to their success in supervised learning settings. This underperformance stems from the unique distribution shifts inherent to offline RL, which increase the sensitivity of parameters and negatively impact performance. Regularization-based methods impose additional constraints to limit parameter updates and maintain stability, as evidenced by their lower forgetting rates compared to Finetuning baseline. However, these methods still experience significant catastrophic forgetting, often exceeding 0.50. Increasing the weight of the regularization loss can reduce forgetting further, but it also hampers the learning of new tasks, exacerbating the stability-plasticity tradeoff (Appendix G). Moreover, tuning these hyperparameters is often challenging and time-consuming, particularly in offline settings.

For rehearsal-based methods, despite their effectiveness in supervised learning, methods like Perfect Memory and A-GEM exhibit poor performance in offline RL, even when allowed a generous replay buffer. We hypothesize that this is due to the additional distribution shift introduced between the replay buffer and the learned policy, as discussed in Gai et al. (2023).

Table 1: Evaluation results (mean ± standard deviation) across three metrics averaged over three random seeds on the Offline Continual World benchmark. MT = Multi-task, Reg = Regularization-based, Struc = Structure-based, Reh = Rehearsal-based, P = Average Performance, F = Forgetting, FWT = Forward Transfer. Detailed descriptions of baselines and metrics are provided in Section 5.1. The top two results among the continual learning methods for each metric are highlighted.

| Benchmarks | | OCW10 | | | OCW20 | | |
|---|---|---|---|---|---|---|---|
| Metrics | | P (↑) | F (↓) | FWT (↑) | P (↑) | F (↓) | FWT (↑) |
| MT | MTL | $0.75 \pm 0.04$ | - | - | $0.80 \pm 0.02$ | - | - |
| Reg | L2 | $0.29 \pm 0.06$ | $-0.01 \pm 0.00$ | $0.03 \pm 0.05$ | $0.20 \pm 0.01$ | $0.00 \pm 0.00$ | $0.01 \pm 0.01$ |
| | EWC | $0.16 \pm 0.02$ | $0.67 \pm 0.02$ | $0.02 \pm 0.03$ | $0.12 \pm 0.02$ | $0.69 \pm 0.02$ | $0.05 \pm 0.01$ |
| | MAS | $0.29 \pm 0.04$ | $0.47 \pm 0.07$ | $0.01 \pm 0.01$ | $0.25 \pm 0.02$ | $0.44 \pm 0.02$ | $0.01 \pm 0.02$ |
| | LwF | $0.21 \pm 0.04$ | $0.62 \pm 0.03$ | $0.06 \pm 0.04$ | $0.10 \pm 0.01$ | $0.70 \pm 0.05$ | $0.06 \pm 0.03$ |
| | RWalk | $0.26 \pm 0.03$ | $0.01 \pm 0.01$ | $-0.02 \pm 0.01$ | $0.17 \pm 0.01$ | $0.05 \pm 0.02$ | $0.00 \pm 0.01$ |
| | VCL | $0.14 \pm 0.03$ | $0.68 \pm 0.04$ | $0.04 \pm 0.03$ | $0.06 \pm 0.00$ | $0.74 \pm 0.04$ | $0.04 \pm 0.01$ |
| | Finetuning | $0.11 \pm 0.03$ | $0.73 \pm 0.04$ | $0.03 \pm 0.01$ | $0.08 \pm 0.00$ | $0.77 \pm 0.03$ | $0.03 \pm 0.02$ |
| Struc | LoRA | $0.54 \pm 0.03$ | $0.00 \pm 0.00$ | $0.01 \pm 0.00$ | $0.54 \pm 0.01$ | $0.00 \pm 0.00$ | $0.02 \pm 0.02$ |
| | PackNet | $0.64 \pm 0.06$ | $-0.01 \pm 0.01$ | $0.03 \pm 0.01$ | $0.57 \pm 0.04$ | $0.00 \pm 0.00$ | $0.05 \pm 0.02$ |
| Reh | PM | $0.26 \pm 0.01$ | $0.56 \pm 0.05$ | $0.04 \pm 0.03$ | $0.26 \pm 0.08$ | $0.57 \pm 0.09$ | $0.11 \pm 0.02$ |
| | A-GEM | $0.12 \pm 0.04$ | $0.70 \pm 0.06$ | $0.02 \pm 0.01$ | $0.09 \pm 0.01$ | $0.73 \pm 0.04$ | $0.06 \pm 0.02$ |
| Ours | Grow | $0.60 \pm 0.06$ | $-0.01 \pm 0.01$ | $-0.01 \pm 0.01$ | $0.61 \pm 0.02$ | $0.01 \pm 0.02$ | $-0.01 \pm 0.03$ |
| | Prune | $0.69 \pm 0.01$ | $-0.01 \pm 0.03$ | $0.00 \pm 0.00$ | $0.73 \pm 0.04$ | $0.00 \pm 0.01$ | $0.03 \pm 0.02$ |

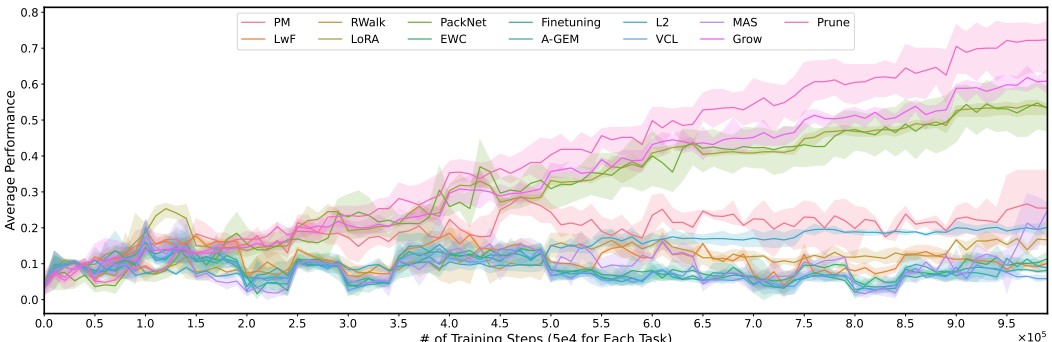

Figure 3: Performance across 3 random seeds for all methods on the OCW20 sequence. CompoFormer-Grow and Prune outperform all baselines, demonstrating faster task adaptation.

Structure-based methods, on the other hand, show better performance, benefiting from parameter isolation that prevents interference between tasks. However, these methods fail to effectively leverage the benefits of transferring relevant knowledge from previous tasks via shared parameters, which impedes the overall learning process. When the second set of tasks in OCW20 is introduced—identical to the first set—the performance of these methods declines, indicating inefficient use of network capacity and prior knowledge. Since no new capacity is required, this suggests that direct parameter sharing without adaptive selection is suboptimal.

In contrast, our method, CompoFormer, addresses these issues by selectively leveraging relevant knowledge from previous policies through its self-composing policy module. This allows for more efficient task adaptation, resulting in faster learning of new tasks (Figure 3) and better performance compared to directly sharing representations. CompoFormer consistently outperforms all other methods in both forgetting (stability) and new task performance (plasticity) across various settings, particularly in the OCW20 benchmark, which demands effective utilization of prior knowledge without incurring capacity limitations. However, we observe that most continual learning methods, including ours, struggle with forward transfer, highlighting the ongoing challenge of improving future tasks' performance by learning new ones. We leave this issue for future work.

## 5.3 ABLATION STUDIES

**Textual description.** Does the output of the attention module in CompoFormer capture the semantic correlations between tasks? To address this question, we visualize the similarity of CompoFormer's attention output between each pair of tasks in the OCW10 sequence, as shown in Figure 4a. Detailed

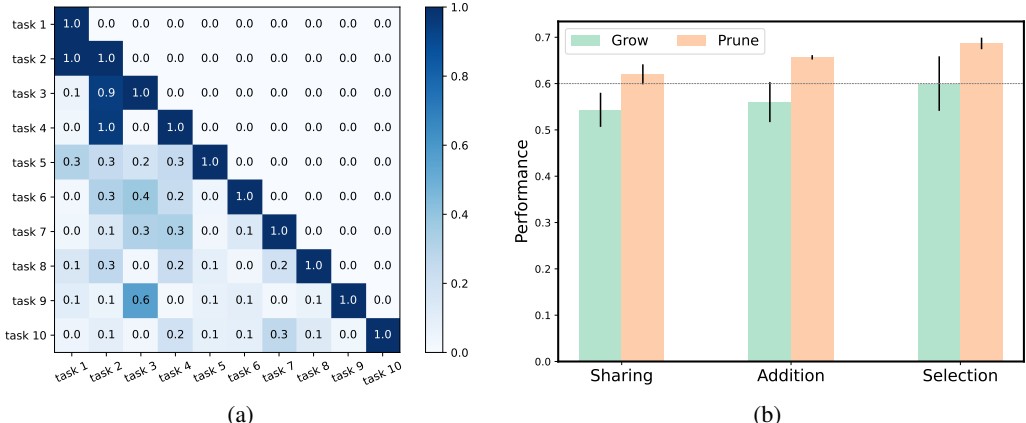

(a)                                             (b)

Figure 4: (a) Visualization of attention scores from the self-composing policy module in the OCW10 benchmark with the CompoFormer-Grow edition, where the diagonal is excluded and set to 1. (b) Evaluation of the effectiveness of our self-composing policy module through three variants: Sharing, Addition, and Selection, conducted in the OCW10 benchmark for both the CompoFormer-Grow and CompoFormer-Prune editions. Each result is averaged over three random seeds.

information regarding the attention scores for the OCW20 sequence can be found in Appendix F. The heatmap illustrates the attention scores assigned to each task based on previous tasks. For example, tasks 2 and 4 share a common manipulation primitive—pushing the puck—reflected in their task descriptions. As a result, when learning the policy for task 4, the model incorporates more knowledge from task 2. In contrast, for tasks with unrelated descriptions, CompoFormer reduces attention to those tasks, thereby minimizing cross-task interference and enhancing plasticity.

**Effectiveness of design choices.** To demonstrate the effectiveness of our self-composing policy module, we conduct an ablation study on three variants of CompoFormer, each altering a single design choice from the original framework. We evaluate the following variants: (1) Layer-Sharing: directly shares the layer representations output by each hidden layer without any selection mechanism; (2) Direct-Addition: directly adds the output of previous policies to the current policy; (3) Attentive-Selection: utilizes an attention mechanism to selectively incorporate the most relevant previous policies. As illustrated in Figure 4b, incorporating previous knowledge at the policy level, rather than through direct sharing of hidden layer representations, leads to improved performance. Additionally, the Attentive-Selection variant further enhances the benefits of knowledge sharing by minimizing cross-task interference and improving plasticity.

**Impact of sequence order.** Considering the influence of sequence order on continual learning performance (Singh et al., 2023; Zhou et al., 2023), we use different random seeds to shuffle the task sequences and conduct experiments on OCW10 (Detailed in Appendix A). The average performance of these methods is presented in Table 2. For most regularization- and rehearsal-based methods, changes in task sequence order result in only minor performance variations; however, overall performance remains poor. This suggests that distributional shift continues to play a dominant role, overshadowing the effect of task order. For structure-based methods, the impact of task order is more pronounced. In the case of LoRA, which trains the full model parameters on the first task and only fine-tunes the LoRA-Linear parameters for subsequent tasks, the first task significantly influences overall performance, leading to substantial performance variation. Conversely, PackNet, which assigns separate parameters to each task, shows minimal sensitivity to task order changes. Our method, which attentively selects prior policies, may exhibit varying degrees of knowledge transfer depending on task sequence order. However, it consistently outperforms LoRA and PackNet, demonstrating robustness to task sequence variations. This indicates that our approach effectively selects the most relevant knowledge from previously learned policies, regardless of sequence order.

**Plasticity.** Structure-based methods often maintain stability by fixing learned parameters after completing a task and directly sharing layer-wise hidden representations. However, this approach may compromise plasticity, the ability to learn new tasks effectively. To evaluate whether our method sacrifices plasticity, we conduct ablation studies by assessing the performance of each task in a single-task setting, using the same network parameters to evaluate the model's performance on individual tasks. The single-task performance in the OCW10 setting is then analyzed to determine

Table 2: Average performance (mean ± standard deviation) across three random seeds for different task orders in the OCW10 benchmark. Reg = Regularization-based, Struc = Structure-based, Reh = Rehearsal-based. The top two results are highlighted. "Order 0" refers to the original task order, while "Order 1", "Order 2", and "Order 3" represent random shuffles using seeds 1, 2, and 3.

| Benchmarks | | OCW10 | | | | |
|---|---|---|---|---|---|---|
| Order | | 0 | 1 | 2 | 3 | Average |
| Reg | L2 | $0.29 \pm 0.06$ | $0.20 \pm 0.03$ | $0.23 \pm 0.05$ | $0.29 \pm 0.05$ | 0.25 |
| | EWC | $0.16 \pm 0.02$ | $0.12 \pm 0.02$ | $0.11 \pm 0.02$ | $0.15 \pm 0.03$ | 0.13 |
| | MAS | $0.29 \pm 0.04$ | $0.17 \pm 0.01$ | $0.16 \pm 0.06$ | $0.20 \pm 0.04$ | 0.21 |
| | LwF | $0.21 \pm 0.04$ | $0.15 \pm 0.04$ | $0.18 \pm 0.05$ | $0.19 \pm 0.00$ | 0.18 |
| | RWalk | $0.26 \pm 0.03$ | $0.17 \pm 0.01$ | $0.25 \pm 0.02$ | $0.20 \pm 0.01$ | 0.22 |
| | VCL | $0.14 \pm 0.03$ | $0.11 \pm 0.01$ | $0.11 \pm 0.01$ | $0.12 \pm 0.03$ | 0.12 |
| | Finetuning | $0.11 \pm 0.03$ | $0.12 \pm 0.02$ | $0.15 \pm 0.06$ | $0.12 \pm 0.03$ | 0.12 |
| Struc | LoRA | $0.54 \pm 0.03$ | $0.47 \pm 0.02$ | $0.35 \pm 0.03$ | $0.43 \pm 0.05$ | 0.45 |
| | PackNet | $0.64 \pm 0.06$ | $0.67 \pm 0.01$ | $0.65 \pm 0.03$ | $0.65 \pm 0.04$ | 0.65 |
| Reh | PM | $0.26 \pm 0.01$ | $0.26 \pm 0.10$ | $0.25 \pm 0.03$ | $0.27 \pm 0.01$ | 0.26 |
| | A-GEM | $0.12 \pm 0.04$ | $0.12 \pm 0.02$ | $0.11 \pm 0.01$ | $0.15 \pm 0.04$ | 0.13 |
| Ours | Grow | $0.60 \pm 0.06$ | $0.54 \pm 0.05$ | $0.43 \pm 0.01$ | $0.51 \pm 0.05$ | 0.52 |
| | Prune | $0.69 \pm 0.01$ | $0.70 \pm 0.03$ | $0.70 \pm 0.03$ | $0.68 \pm 0.02$ | 0.69 |

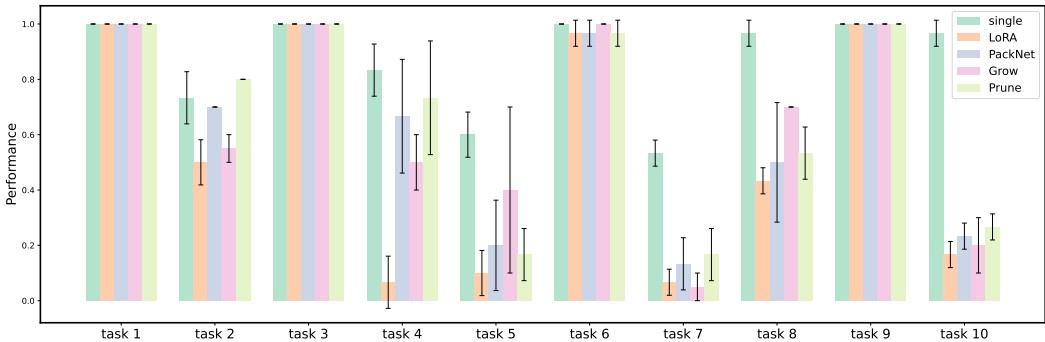

Figure 5: Performance across three random seeds for each task in the OCW10 benchmark. "Single" refers to the performance of individual task training, while the other methods reflect each task's performance after the entire learning process.

if any performance degradation has occurred in these continual learning methods (full analysis in Appendix E). As shown in Figure 5, LoRA and PackNet exhibit significant performance drops when compared to single-task evaluations across most tasks, indicating a greater loss in plasticity. While our method also shows some performance reduction, the incorporation of the self-composing policy module results in greater performance benefits compared to the base model. This demonstrates that effectively leveraging prior knowledge can enhance plasticity while still maintaining stability.

## 6 CONCLUSION

We introduce CompoFormer, a modular growing architecture designed to mitigate catastrophic forgetting and task interference while leveraging knowledge from previous tasks to address new ones. In our experiments, we develop the Offline Continual World benchmark to perform a comprehensive empirical evaluation of existing continual learning methods. CompoFormer consistently outperforms these approaches, offering a promising balance between plasticity and stability, all without the need for experience replay or storage of past task data.

We believe this work represents a significant step forward in the development of continual offline reinforcement learning agents capable of learning from numerous sequential tasks. However, challenges remain, particularly concerning the computational cost, which is critical in never-ending continual learning scenarios, and thus warrants further research.

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

## A  OFFLINE CONTINUAL WORLD BENCHMARK

We visualize all tasks in the Offline Continual World benchmark in Figure 6, and provide detailed descriptions in Table 3. The corresponding offline datasets are generated by training a Soft Actor-Critic (SAC) (Haarnoja et al., 2018) policy in isolation for each task from scratch until convergence. Once the policy converges, we collect 1 million transitions from the SAC replay buffer for each task, comprising samples observed during training as the policy approaches optimal performance (Hu et al., 2024c; He et al., 2024).

In the ablation study described in Section 5.3, we use different random seeds to shuffle the task sequences and conduct experiments on OCW10. The specific task sequences for each order are as follows:

- Order 0: [hammer-v2, push-wall-v2, faucet-close-v2, push-back-v2, stick-pull-v2, handle-press-side-v2, push-v2, shelf-place-v2, window-close-v2, peg-unplug-side-v2]

- Order 1: [push-v2, window-close-v2, peg-unplug-side-v2, shelf-place-v2, handle-press-side-v2, push-back-v2, hammer-v2, stick-pull-v2, push-wall-v2, faucet-close-v2]

- Order 2: [handle-press-side-v2, peg-unplug-side-v2, push-back-v2, stick-pull-v2, push-v2, shelf-place-v2, faucet-close-v2, window-close-v2, push-wall-v2, hammer-v2]

- Order 3: [push-wall-v2, handle-press-side-v2, push-v2, hammer-v2, peg-unplug-side-v2, stick-pull-v2, shelf-place-v2, faucet-close-v2, window-close-v2, push-back-v2]

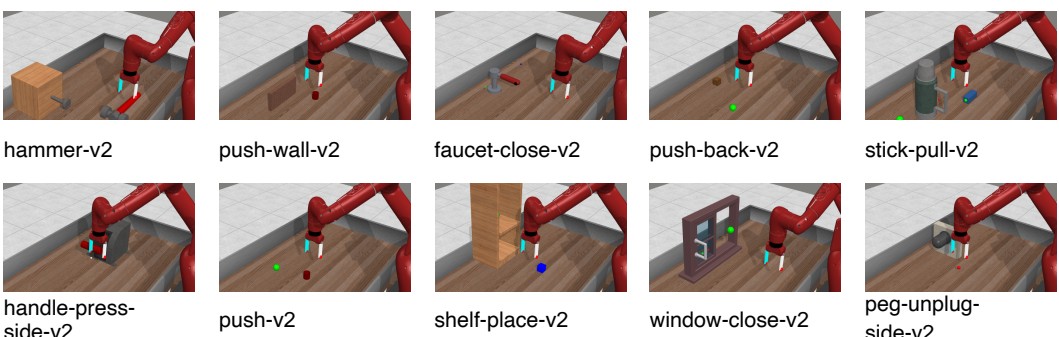

Figure 6: The Offline Continual World benchmark comprises robotic manipulation tasks from Meta-World (Yu et al., 2020). The OCW10 sequence shown above is used in the main experimental results.

Table 3: A list of all tasks in the Offline Continual World benchmark along with a description of each. The OCW10 sequence is shown below, while the OCW20 sequence consists of the same tasks repeated twice. Tasks are learned sequentially, with a maximum of 5e5 training iterations per task.

| Index | Task | Description |
|---|---|---|
| 1 | hammer-v2 | Hammer a screw on the wall. |
| 2 | push-wall-v2 | Bypass a wall and push a puck to a goal. |
| 3 | faucet-close-v2 | Rotate the faucet clockwise. |
| 4 | push-back-v2 | Pull a puck to a goal. |
| 5 | stick-pull-v2 | Grasp a stick and pull a box with the stick. |
| 6 | handle-press-side-v2 | Press a handle down sideways. |
| 7 | push-v2 | Push the puck to a goal. |
| 8 | shelf-place-v2 | Pick and place a puck onto a shelf. |
| 9 | window-close-v2 | Push and close a window. |
| 10 | peg-unplug-side-v2 | Unplug a peg sideways. |

# B    AN EXTENDED DESCRIPTION OF COMPARED METHODS

We evaluate 11 continual learning methods on our benchmark: L2, EWC, MAS, LwF, RWalk, VCL, Finetuning, LoRA, PackNet, Perfect Memory, and A-GEM. Most of these methods are originally developed in the context of supervised learning. To ensure comprehensive coverage of different families of approaches within the community, we select methods representing three main categories, as outlined in Masana et al. (2022): regularization-based, structure-based, and rehearsal-based methods. The majority of the methods' implementations are adapted from `https://github.com/mmasana/FACIL`, with their default hyperparameters applied.

## B.1    REGULARIZATION-BASED METHODS

Regularization-based methods focus on preventing the drift of parameters deemed important for previous tasks. These methods estimate the importance of each parameter in the network (assumed to be independent) after learning each task. When training on new tasks, the importance of each parameter is used to penalize changes to those parameters. For example, to retain knowledge from task 1 while learning task 2, a regularization term of the form:

$$\lambda \sum_j F_j (\theta_j - \theta_j^{(1)})^2,\tag{7}$$

is added, where $\theta_j$ are the current weights, $\theta_j^{(1)}$ are the weights after learning the first task, and $\lambda > 0$ is the regularization strength. The coefficients $F_j$ are crucial as they indicate the importance of each parameter. Regularization-based methods are often derived from a Bayesian perspective, where the regularization term constrains learning to stay close to a prior, which incorporates knowledge from previous tasks.

**L2.** This method assumes all parameters are equally important, i.e., $F_j^{(i)} = 1$ for all $j$ and $i$. While simple, L2 regularization reduces forgetting but often limits the ability to learn new tasks.

**EWC.** Elastic Weight Consolidation (EWC) (Kirkpatrick et al., 2017) is grounded in Bayesian principles, treating the parameter distribution of previous tasks as a prior when learning new tasks. Since the distribution is intractable, it is approximated using the diagonal of the Fisher Information Matrix:

$$F_j = \mathbb{E}_{x \sim \mathcal{D}} \mathbb{E}y \sim \pi_\theta(\cdot|x) \left( \nabla_{\theta_j} \log \pi_\theta(y|x) \right)^2.\tag{8}$$

**MAS.** Memory Aware Synapses (Aljundi et al., 2018) estimates the importance of each parameter by measuring the sensitivity of the network's output to weight perturbations. Formally,

$$F_j = \mathbb{E}_{x \sim \mathcal{D}} \left( \frac{\partial ||\pi_\theta(x)||_2^2}{\partial \theta_j} \right),\tag{9}$$

where $\pi_\theta(x)$ is the model output, and the expectation is over the data distribution $\mathcal{D}$.

**LwF.** Learning without Forgetting (Li & Hoiem, 2017) utilizes knowledge distillation to preserve the representations of previous tasks while learning new ones. The training objective includes an additional loss term:

$$\mathbb{E}_{x \sim \mathcal{D}}(\pi_\theta^{(k-1)}(x) - \pi_\theta^{(k)}(x))^2,\tag{10}$$

where $\pi_\theta^{(k-1)}(x)$ and $\pi_\theta^{(k)}(x)$ are the outputs of the old and current models, respectively, and the expectation is over the data distribution $\mathcal{D}$.

**RWalk.** Riemannian Walk (Chaudhry et al., 2018a) generalizes EWC++ and Path Integral (Zenke et al., 2017) by combining Fisher Information-based importance with optimization-path-based scores. The importance of parameters is defined as:

$$F_j = (Fisher_{\theta_j^{(k-1)}} + s_{t_0}^{t_{k-1}}(\theta_j))(\theta_j - \theta_j^{(k-1)})^2,\tag{11}$$

where $s_{t_0}^{t_{k-1}}$ accumulates importance from the first training iteration $t_0$ to the last iteration $t_{k-1}$ for task $k-1$.

**VCL.** Variational Continual Learning (Nguyen et al., 2017) builds on Bayesian neural networks by maintaining a factorized Gaussian distribution over network parameters and applying variational inference to approximate the Bayes update. The training objective includes an additional term:

$$\lambda D_{\text{KL}}(\theta \parallel \theta^{(k-1)}), \tag{12}$$

where $D_{\text{KL}}$ is the Kullback-Leibler divergence and $\theta^{(k-1)}$ represents the parameter distribution after learning the previous tasks.

## B.2 STRUCTURE-BASED METHODS

Structure-based methods, also referred to as modularity-based methods, preserve previously acquired knowledge by keeping specific sets of parameters fixed. This approach imposes a hard constraint on the network, in contrast to the soft regularization penalties used in regularization-based methods.

**LoRA.** As introduced by Huang et al. (2024), this method adds a new LoRA-Linear module when a new task is introduced. According to Lawson & Qureshi (2024), in sequential decision-making tasks, Decision Transformers rely more heavily on MLP layers than on attention mechanisms. LoRA leverages this by merging weights that contribute minimally to knowledge sharing and fine-tuning the decisive MLP layers in DT blocks with LoRA to adapt to the current task.

**PackNet.** Introduced by Mallya & Lazebnik (2018), this method iteratively applies pruning techniques after each task is trained, effectively "packing" the task into a subset of the neural network parameters, while leaving the remaining parameters available for future tasks. The parameters associated with previous tasks are frozen, thus preventing forgetting. Unlike earlier methods, such as progressive networks (Rusu et al., 2016), PackNet maintains a fixed model size throughout learning. However, the number of available parameters decreases with each new task. Pruning in PackNet is a two-stage process. First, a fixed subset of the most important parameters for the task is selected, typically comprising 70% of the total. In the second stage, the network formed by this subset is fine-tuned over a specified number of steps.

## B.3 REHEARSAL-BASED METHODS

Rehearsal-based methods mitigate forgetting by maintaining a buffer of samples from previous tasks, which are replayed during training.

**Perfect Memory.** This method assumes an unrealistic scenario of an unlimited buffer capacity, allowing all data from previous tasks to be stored and replayed.

**A-GEM.** Averaged Gradient Episodic Memory (Chaudhry et al., 2018b) formulates continual learning as a constrained optimization problem. Specifically, the objective is to minimize the loss for the current task, $\ell(\theta, \mathcal{D}^{(k)})$, while ensuring that the losses for previous tasks remain bounded, $\ell(\theta, \mathcal{M}^{(i)}) \leq \ell_i$, where $\ell_i$ represents the previously observed minimum, and $\mathcal{M}^{(i)}$ contains buffer samples from task $i$ for $1 \leq i \leq k-1$. However, this constraint is intractable for neural networks. To address this, Chaudhry et al. (2018b) propose an approximation using a first-order Taylor expansion:

$$\langle \nabla_\theta \ell(\theta, \mathcal{B}_{new}), \nabla_\theta \ell(\theta, \mathcal{B}_{old}) \rangle > 0, \tag{13}$$

where $\mathcal{B}_{new}$ and $\mathcal{B}_{old}$ represent batches of data from the current and previous tasks, respectively. This constraint is implemented via gradient projection:

$$\nabla_\theta \ell(\theta, \mathcal{B}_{new}) - \frac{\langle \nabla_\theta \ell(\theta, \mathcal{B}_{new}), \nabla_\theta \ell(\theta, \mathcal{B}_{old}) \rangle}{\langle \nabla_\theta \ell(\theta, \mathcal{B}_{old}), \nabla_\theta \ell(\theta, \mathcal{B}_{old}) \rangle} \nabla_\theta \ell(\theta, \mathcal{B}_{old}). \tag{14}$$

## C HYPERPARAMETER DETAILS AND RESOURCES

**Training Details**. To ensure fairness and reproducibility in the presented experiments, all continual learning methods are implemented using the Decision Transformer (Chen et al., 2021) architecture with identical hyperparameters for the architectural components, as detailed in Table4.

Table 4: Hyperparameters of Transformer in our experiments.

| Parameter | Value |
|---|---|
| Number of layers | 6 |
| Number of attention heads | 8 |
| Embedding dimension | 256 |
| Nonlinearity function | ReLU |
| Batch size | 32 |
| Context length $K$ | 20 |
| Dropout | 0.1 |
| Learning rate | 1.0e-4 |
| Per task iterations | 5e4 |
| Performance threshold $\eta$ | 0.8 |

**Training Resources**. All methods are trained using an NVIDIA GeForce RTX 4090 GPU. The training duration varies by method. For example, on the OCW10 benchmark, regularization-based and rehearsal-based methods typically require 14 to 18 hours, while structure-based methods, which frequently involve saving and loading model parameters, take approximately 3 to 4 days. Since each environment is trained three times with different random seeds, the total training time is approximately three times the duration of a single run.

## D    COMPUTATIONAL COST OF INFERENCE IN COMPOFORMER

In this section, we analyze the computational cost of inference in the CompoFormer architecture with respect to the number of tasks, expressing the complexity in big-$O$ notation.

As shown in Figure 1 and Algorithm 1, the primary computational cost during inference arises from the self-composing policy module. In the worst-case scenario, where no task can be solved by directly composing previous policies, the main inference time for a given task can be simplified as the cost of attention calculation over the matrix of previous policy outputs, $\Phi^{(1:k-1)}$. This matrix has dimensions $(k-1) \times h$, where $h$ is constant with respect to the number of modules, but the first dimension grows linearly with the number of tasks, $k$. Let $d$ represent the time cost per operation, the time required to construct $\Phi^{(1:k-1)}$ (denoted as matrix $\boldsymbol{V}$) is given by:

$$T_{\Phi^{(1:k-1)}}(k) = (k-1) \cdot d_{\text{model}}. \tag{15}$$

The computational cost of calculating the key matrix $\boldsymbol{K}$ and the subsequent dot-product operation for attention depends linearly on the number of policies, $k$.[1] Let $h$ denote the hidden dimension. The time complexity for the attention module, $T_{\text{attn}}(k)$, can be expressed as:

$$T_{\text{attn}}(k) = \underbrace{d_{\text{enc}} \cdot d_{\text{linear}}}_{\text{Compute q}} + \underbrace{(k-1) \cdot d_{\text{enc}} \cdot d_{\text{linear}}}_{\text{Compute K}} + \underbrace{(k-1) \cdot h}_{\boldsymbol{q}\boldsymbol{K}^T} + \underbrace{O(k-1)}_{\text{Cost of softmax}} + \underbrace{(k-1) \cdot h}_{\text{Mult. att. and V}} . \tag{16}$$

Since each task introduces new parameters that output features, which are combined with previous output features to construct the final action output, the total time complexity is:

$$T(k) = T_{\Phi^{(1:k-1)}}(k) + T_{\text{attn}}(k) + d_{model}. \tag{17}$$

Given that there are no higher-order terms beyond $k$, the computational complexity for the inference operation of a single module is $T(k) = O(k)$.

Given a CompoFormer model consisting of $k$ policy modules, generating the final output requires sequentially computing the results of all $k$ modules, as each module's output depends on the results of the preceding ones. Consequently, although the complexity of inference for each individual module is linear, the overall complexity of inference in CompoFormer is $k \cdot O(k) = O(k^2)$.

---

[1]The computational complexity of multiplying an $n \times m$ matrix by an $m \times p$ matrix is $O(nmp)$.

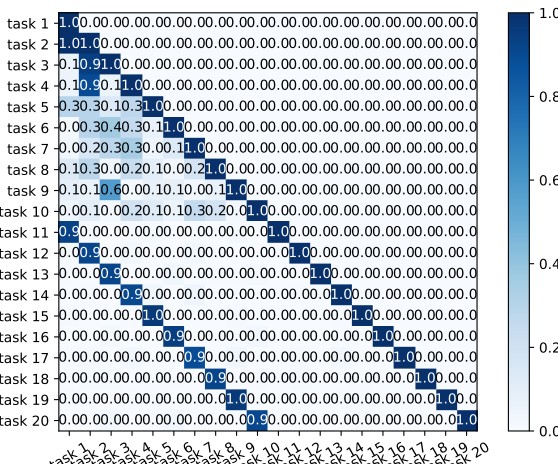

Figure 7: Visualization of attention scores from the self-composing policy module in the OCW20 benchmark with the CompoFormer-Grow edition, where the diagonal is excluded and set to 1.

The computational cost remains a primary limitation in growing neural network architectures, with memory and inference costs being key considerations in this study. The self-composing policy module in CompoFormer is designed to address memory complexity by ensuring that the model grows linearly in terms of parameters as the number of tasks increases, while still retaining the knowledge from all previously learned modules. Furthermore, when a new task can be solved by reusing previous policies, no additional computational cost for training is incurred, significantly reducing the computational burden.

## E    DETAILED SINGLE PERFORMANCE IN OUR BASELINES

In Section 5.3, we visualized the single-task performance of five methods; here, we present the performance of all methods in Figure 8. As shown in Figure 8, the MT method, which has access to data from all tasks throughout the learning process, achieves high performance on every task, comparable to the single-task method. In contrast, regularization-based and rehearsal-based methods introduce additional loss terms to mitigate catastrophic forgetting. However, during training, these methods still experience forgetting, particularly for the initial tasks in the sequence, and only show improved performance towards the end of the learning sequence. Increasing the regularization strength to prevent forgetting results in impaired learning of new tasks, leading to a pronounced stability-plasticity trade-off. This issue is particularly significant in the offline RL setting, where policy optimization is more sensitive to parameter changes compared to supervised learning in classification tasks(Masana et al., 2022; Zhou et al., 2023).

Structure-based methods maintain good stability by freezing task-specific parameters, but their ability to learn new tasks diminishes as the training progresses. This is due to the decreasing number of available parameters and cross-task interference from the frozen parameters. Consequently, the performance on later tasks is often lower than that of regularization- and rehearsal-based methods. Our self-composing policy module addresses this issue by reducing cross-task interference and enhancing the plasticity of structure-based methods, allowing for more efficient learning of new tasks without compromising stability.

## F    DETAILED ATTENTION SCORES IN OCW20

In Section 5.3, we visualized the attention scores for the OCW10 benchmark. Here, we extend this analysis by visualizing the attention scores for the OCW20 benchmark, which repeats the OCW10 task sequence twice. As shown in Figure 7, for the first 10 tasks, the attention scores are similar to those observed in the OCW10 benchmark. However, for the subsequent 10 tasks, the model assigns the highest attention to the corresponding tasks from the first sequence, which are most relevant to the current task. This demonstrates the model's ability to capture semantic correlations between tasks and effectively leverage previously learned policies, resulting in superior performance compared to the baselines.

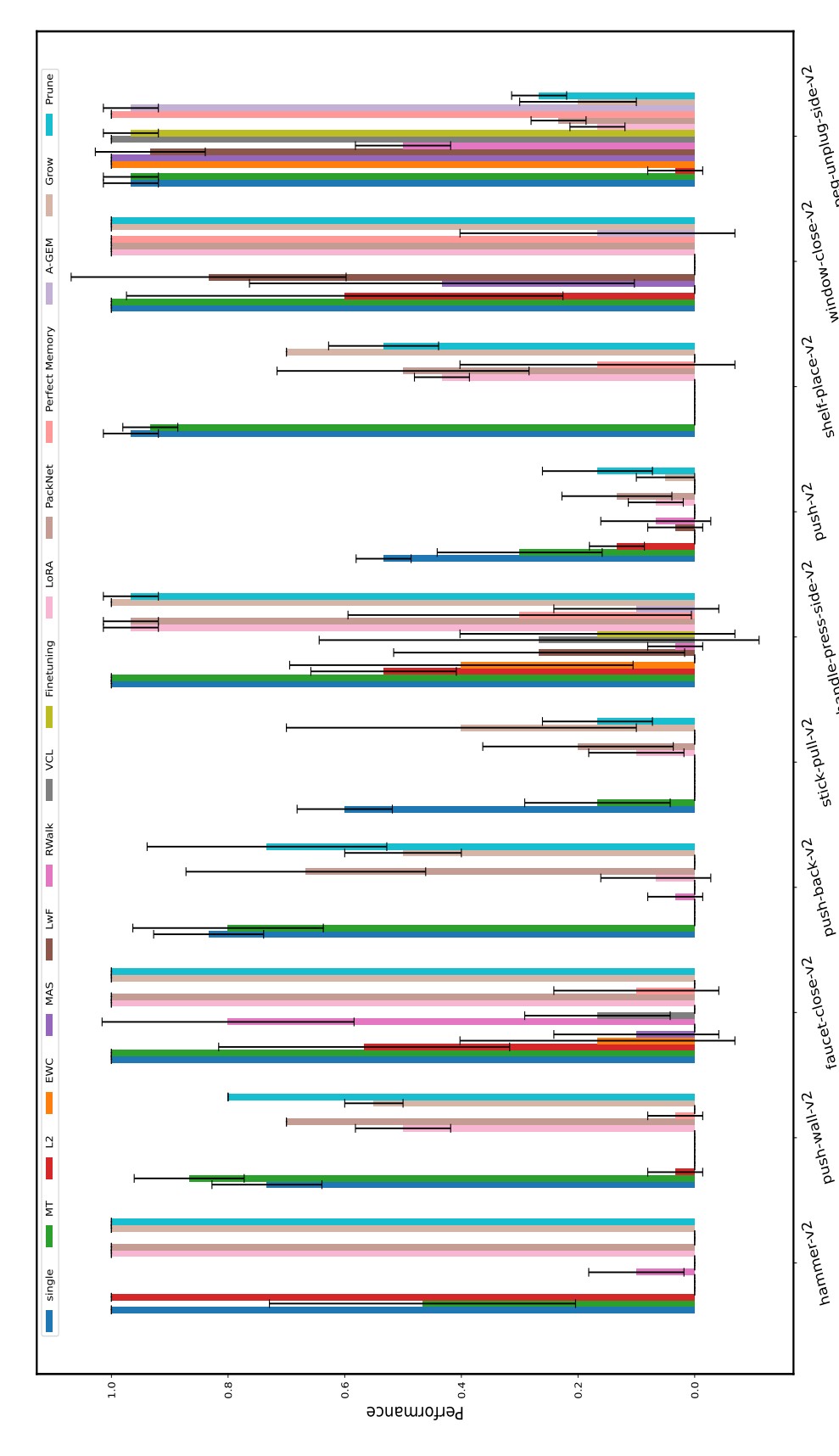

Figure 8: Performance across three random seeds for each task in the OCW10 benchmark, evaluated across all methods. "Single" denotes the performance of individual task training, while the other methods represent each task's performance after completing the entire learning process.

## G  HYPERPARAMETERS OF BASELINES

This section outlines the hyperparameters utilized in each baseline, with particular emphasis on the weight of the regularization term, as detailed in Section B.

We test the following hyperparameter values: $\lambda \in \{10^{-1}, 10^0, 10^1, 10^2, 10^3, 5 \times 10^3, 10^4\}$, and the results are presented in Table 5. The hyperparameters corresponding to the best performance are employed in the main results. Notably, we observe no direct correlation between the model's final performance and the hyperparameters, indicating that tuning these settings often requires substantial manpower and resources.

Table 5: Average performance (mean ± standard deviation) across three random seeds in the OCW10 benchmark, evaluated with varying hyperparameters.

| $\lambda$ | $10^{-1}$ | $10^0$ | $10^1$ | $10^2$ | $10^3$ | $5 \times 10^3$ | $10^4$ |
|---|---|---|---|---|---|---|---|
| L2 | $0.18 \pm 0.03$ | $0.20 \pm 0.03$ | $0.21 \pm 0.04$ | $0.22 \pm 0.03$ | $0.25 \pm 0.05$ | $0.29 \pm 0.06$ | $0.25 \pm 0.05$ |
| EWC | $0.13 \pm 0.03$ | $0.14 \pm 0.03$ | $0.10 \pm 0.03$ | $0.15 \pm 0.03$ | $0.11 \pm 0.05$ | $0.16 \pm 0.02$ | $0.14 \pm 0.02$ |
| MAS | $0.12 \pm 0.02$ | $0.15 \pm 0.03$ | $0.29 \pm 0.04$ | $0.18 \pm 0.04$ | $0.13 \pm 0.02$ | $0.10 \pm 0.01$ | $0.15 \pm 0.03$ |
| LwF | $0.18 \pm 0.02$ | $0.21 \pm 0.04$ | $0.13 \pm 0.03$ | $0.20 \pm 0.02$ | $0.17 \pm 0.03$ | $0.20 \pm 0.02$ | $0.15 \pm 0.03$ |
| RWalk | $0.13 \pm 0.04$ | $0.15 \pm 0.02$ | $0.15 \pm 0.03$ | $0.11 \pm 0.04$ | $0.26 \pm 0.03$ | $0.19 \pm 0.02$ | $0.21 \pm 0.03$ |
| VCL | $0.12 \pm 0.03$ | $0.14 \pm 0.03$ | $0.06 \pm 0.03$ | $0.03 \pm 0.04$ | $0.07 \pm 0.03$ | $0.12 \pm 0.02$ | $0.11 \pm 0.02$ |

