# OpenReview forum: "Continual Task Learning through Adaptive Policy Self-Composition"
_ICLR.cc/2025/Conference — ICLR 2025 Conference Withdrawn Submission_

### Official Review · Reviewer_z4q8 · 2024-10-28

**Soundness:** 4
**Presentation:** 4
**Contribution:** 2
**Rating:** 6
**Confidence:** 3

**Summary:**

The authors explore the problem of offline continual RL by introducing a benchmark based on a prior online continual RL benchmark and a novel method.  Their method balances plasticity and stability by using an attention mechanism with textual task descriptions, a Decision Transformer backbone for the policy, and two different methods of growing or pruning the policy parameters in order to learn new tasks.  They provide an extensive comparison on their benchmark, demonstrating that online continual RL algorithms struggle in this setting, and strong performance for both versions of their method.

**Strengths:**

* The authors propose a sound and flexible continual RL method.
* Strong experimental results and analysis.
* Clarity of writing

**Weaknesses:**

My only concern is on the importance of the continual offline RL problem, instead of a more general multi-task offline RL problem.  It seems like doing continual learning is just imposing an artificial constraint on what order the agent sees the data, one that the “Prune” version of this method is actually violating when it re-trains on old tasks.  Meanwhile doing multi-task offline RL with the same data assumptions can result in more effective algorithms that do not have to deal with catastrophic forgetting at all.

**Questions:**

* In Section 4.2, it’s unclear how $\{W^Q, W^K\}$ are updated and what the pre-defined threshold is on.  Is the policy always defined by Eqn. 6 or is that only the case when the architecture is expanded?
* Plasticity analysis experiment. The paper says “LoRA and PackNet exhibit significant performance drops” while your method “results in greater performance benefits compared to the base model” implying your method outperforms the comparison methods.  I don’t see clear evidence of this for PackNet in Figure 5.

---

### Official Review · Reviewer_zALG · 2024-11-03

**Soundness:** 3
**Presentation:** 2
**Contribution:** 2
**Rating:** 5
**Confidence:** 3

**Summary:**

The paper has two contributions:

1. Continual Offline RL benchmark based on existing Continual World benchmark

2. A method to solve this benchmark which relies on composing pre-trained policies through attention mechanism and optionally learning new policy which can be added to the set of the policies

**Strengths:**

1. Novel method to combine policies from a fixed set of policies through attention mechanism, as well as the way to optionally add new policies to this set by learning LoRA parameters for the base model (transformer).

Overall, using attention-based mechanism for combining policies is promising and novel. There is already evidence that it is a very effective way to combine knowledge (see, for example Flamingo paper (https://arxiv.org/abs/2204.14198) for how attention-based mechanism is used to combine different modalities together). Also, the fact that the proposed method can grow the set of policies over time is promising, since it can allow for more effective learning in future situations.

2. Experiments which demonstrate that the proposed method is effective compared to existing baselines. Overall, the signal seems clear, but more clarifications are required (see Weaknesses and Questions sections).

**Weaknesses:**

# Major weakness:

## Why Continual Offline Reinforcement Learning ?

The paper lacks a clear discussion / motivation for the considered Continual Offline Reinforcement Learning setting. It mainly postulates that this setting is important or that successfully solving this is a natural requirement for a long-lived agents. However, it is unclear why one should care about this setting. In particular, if one has access to a sequence of offline dataset, a natural thing to do would be to just combine them in a one large dataset and train transformer i.i.d. Having a sequential set of dataset is not well motivated.
One specific setting I can imagine, which makes sense, is the one where we could pretrain a transformer on a large **offline** dataset and then deploy it in new, **online**, situations. In these new situations, we want our algorithm to maintain plasticity/stability trade-off.

However, this specific setting is not studied in this paper. For example, the authors could have added a baseline of offline pretraining the transformer (their method) and then learn **online** on a sequence of tasks. In fact, as the method is formulated, it is not fully clear why it should be purely offline method.

If the proposed method cannot be applied in the **online** setting, then the authors should explain why and clearly show it in the paper.

## Doubts about experimental validity (I let the authors clarify this in their rebuttal) and lacks of clarity in some important points.

**(a)**

In lines 372-374, the authors say that for regularization-based baselines, tuning the hyperparameters is challenging and time-consuming. The authors do not expand on this, they do not give an example on why it is challenging and time-consuming. From the naive view, say, if I take EWC baseline with regularization strength, lambda, why is it time-consuming and challenging to do a grid search over this parameter?

More importantly, the proposed method by the authors has a parameter, called "performance threshold". The authors do not at all say how they select this parameter, they don't even specify what value they have selected. This parameter since important, especially because one could expect the performance threshold to depend heavily on each given task (imagine just if the reward function has different scales). The authors must include the discussion on how this parameter is selected and how sensitive is the performance of their method to the choice of their parameter.

**(b)**

Another question which is un-addressed -- is the base model which the authors use, pretrained? Section 4, Figure 2 and Algorithm 1 suggest that we do the following:
* We update attention parameters, attending to previous k policies
* We optionally train LoRA parameters (from base parameters of transformer) for new policy

I fail to grasp where do the "base" parameters required for LoRA, are coming from. Is it the case that the transformer weights are randomly initialized and the authors fine-tune LoRA parameters on randomly initialized transformer, while keeping these random parameters in memory for the entire duration of the training? I think the authors should highlight this aspect more in Algorithm 1.

** (c) **

The authors claim that their method is not doing well in terms of "Forward Transfer". This is quite surprising since the form of the method would suggest the opposite, especially if new tasks are similar. Why is it the case?

# Minor weakness:

## Paper presentation can be improved:

* Figure 1 -- it is not clearly explained in figure caption nor in the text, what "enough"/"sufficient" means (it gets slightly clearer later on, but at this stage it is not clear)
* Few related works are not mentioned "Progressive Networks" and "Progress & Compress". I think these works are earlier versions of "structure-based" and "regularized-based" approaches which are relevant for this paper.
* In line 183, it is written "implying that tasks should have similar state spaces, i.e. ...". In fact, if one reads the sentence, it is not at all implied from it. It is in fact just an assumption, the authors impose. This should be re-written and clearly stated.
* In line 196, the authors refer to Algorithm 1, but it is much further away in the text. They should move it so that it appears earlier.
* In line 360, the authors say that MTL and Finetuning (FT) are typically regarded as the soft upper bound. First of all, this is not true that FT and MTL are upper bounds on performance. Second of all, it is not clear what it means by "soft upper bound". The authors should clarify this.
* Figure 3 is very hard to read, especially because the colors of some methods are quite similar to each other. Two optional suggestions: use different line-styles, present less methods in the main paper (one per baseline type) and present the rest in the appendix.

## Some clarity is required on:

* Figure 3 does not contain performance of MTL. MTL can be reported either as a dotted line (trained on all the tasks), or as incremental-MTL, i.e., one trains in the multi-task way on all the K encountered tasks.
* In lines 375-377, the authors claim that rehearshal-based approaches do not work in their setting due to the distribution shift between replay buffer and learned policy. But this is only true for online RL setting, why would this be a problem for the offline RL?

**Questions:**

1. Why do we care about the proposed CORL setting? (see my point in "Weaknesses" section)

2. Can the method be applied in the online RL setting and if not, why?

3. How are the hyperparameters for the proposed method are selected and what is the sensitivity of the method to these parameters?

4. Is the transformer pre-trained before doing CompoFormer? If yes, is it taken into account for the baselines?

5. What is the performance of the MTL / incremental-MTL?

6. Why does distribution shift between replay buffer and learned policy affects rehearshal-based approaches?

7. Why does the proposed method not perform well in terms of forward transfer?

---

### Official Review · Reviewer_HkGj · 2024-11-04

**Soundness:** 3
**Presentation:** 3
**Contribution:** 2
**Rating:** 3
**Confidence:** 4

**Summary:**

Introduces a lifelong offline RL problem statement, where the agent encounters a sequence of tasks with offline data is provided per task. Main contributions are a benchmark, Offline Continual World (based on meta-world tasks), and a method, CompoFormer.

**Strengths:**

Strengths:
- Sequential learning settings are interesting and important to study
- Method is a combination of some nice ideas including low-rank adaptation and leveraging task strings
- Lots of comparisons to prior works in the experiments

**Weaknesses:**

For the problem setting & benchmark
1. One of the assumptions seems unrealistic — The problem setting seems to assume that datasets from previous tasks cannot be accessed at all. It seems unrealistic that you wouldn’t be able to store any of it. For example, many large scale ML systems seem to be more limited by compute than hard drive space, and if there is so much data that it doesn’t fit on a hard drive, then it makes sense to remove some of the data from the current task to accommodate data from previous tasks
2. (Not being able to collect online data from previous tasks seems realistic though)
3. Creating offline datasets by taking the replay buffer of an online agent also lacks some realism. (Perhaps there is a setting where one would run an online RL method to collect data for a new task? But then if that were the case, it wouldn’t make senes to discard some of the samples from the buffer, and it would make sense for the performance of this online agent to be considered as a point of comparison.) While collecting offline benchmark datasets from buffers is a common practice, more recent works (e.g. this paper: https://arxiv.org/pdf/2408.08441) have discussed how it is not representative of real-world challenges.

Experiments:
1. It’s disappointing that forward transfer is essentially 0. It suggests that the tasks might be too different from each other.
2. Single task training seems to significantly outperform the proposed method according to fig 5. How does the parameter count of the proposed method compare to that of single task training? Given the lack of forward transfer in the proposed method and the much greater simplicity of single-task training, it seems like single task learning is actually the best method to use. It involves more parameters than the proposed method, but it also performs substantially better (and is far simpler)
3. I believe the proposed method is the only one that uses online samples to evaluate a version of the policy (i.e. line 14 of the algo box). Is that correct? If so, this comparisons are technically a bit apples-to-oranges / it’s also not 100% offline. How many online samples are used to evaluate?
4. Which of the 10 tasks does the method introduce new parameters for?
5. Ablations would be valuable to understand the contribution of different aspects of the method.


Minor comments:
1. Does it make sense to use the original meta-world task names? (which were not designed for this use-case) I expect it would work better with more descriptive text, e.g. something like “press the side-facing handle downward” instead of “handle press side”
2. It would be nice to evaluate on more tasks. e.g. perhaps tasks/environments from composuite: https://github.com/Lifelong-ML/offline-compositional-rl-datasets?tab=readme-ov-file
3. It would be valuable to compare to prior CORL methods, both in the RW section and experimentally. Right now, the related work does not describe whether they are applicable to the problem setting, and it also doesn’t discuss how they differ technically
4. It would also be valuable to the reader to discuss how compoformer is similar to and differs from prior works at a technical level for works that don’t do CORL. e.g. describing the similarities and differences to progressive nets, progress & compress (even those papers consider an online setting), and papers that do lifelong learning with LoRA (e.g. https://arxiv.org/pdf/2311.17601)
5. These two statements seem contradictory:
    - “Transformer-based methods help reduce distribution shift between the behavior and learned policies”
    - “they may exacerbate shifts […] between the learned policy and replay buffer”

**Questions:**

See questions in the weaknesses section.

---

### Official Review · Reviewer_nUmo · 2024-11-04

**Soundness:** 2
**Presentation:** 3
**Contribution:** 2
**Rating:** 5
**Confidence:** 3

**Summary:**

This paper introduces CompoFormer, a continual learning framework for offline reinforcement learning that enables agents to retain knowledge across sequential tasks while minimizing catastrophic forgetting. Using a modular transformer architecture, CompoFormer selectively composes previous policies, accelerating adaptation to new tasks and balancing plasticity and stability.

**Strengths:**

1. the paper is well written.
2. CompoFormer is capable of retaining knowledge across tasks, mitigating catastrophic forgetting effectively.

**Weaknesses:**

1. The reliance on textual descriptions for attention scores may not generalize well to domains lacking explicit task semantics.
2. It introduces extra training costs, especially for evaluating the ability of the composed policy to solve the current task, i.e., the performance comparison between policies.

**Questions:**

1. I saw some prior literature have explicitly visualized the plasticity of NN [1-3]. Can this work also try to further enhance the claim and contribution?

[1] http://arxiv.org/abs/2205.07802
[2] http://arxiv.org/abs/2204.09560
[3] http://arxiv.org/abs/2305.15555

---

### Note · Authors · 2024-11-20

I have read and agree with the venue's withdrawal policy on behalf of myself and my co-authors.